# OpenReview forum: "Reward Models Inherit Value Biases from Pretraining"
_ICLR.cc/2026/Conference — ICLR 2026 Poster_

### Official Review · Reviewer_ffKc · 2025-10-17

**Soundness:** 4
**Presentation:** 4
**Contribution:** 3
**Rating:** 8
**Confidence:** 4

**Summary:**

This paper investigates whether reward models (RMs) inherit value biases from their pretrained language model backbones. The authors conduct a systematic analysis across 10 open-weight RMs based on either Llama or Gemma architectures and show that these models consistently differ along established psycholinguistic dimensions of human value—particularly the "Big Two" axes of agency and communion. They demonstrate that Llama-based RMs exhibit stronger preferences for agency-related words (e.g., freedom, ability, success), whereas Gemma-based RMs prefer communion-related words (e.g., love, friendship, care). Importantly, the paper traces these differences back to the log-probability structure of the base models themselves and formulates an “implicit reward model” defined by log-probability deltas between two LLMs that captures such differences. Experiments with RM training further show that these biases persist even after training RMs on large preference datasets.

**Strengths:**

- **Relevance and potential impact**: The work tackles an important and underexplored problem in the alignment literature—understanding how pretraining choices influence downstream reward models. Even though the result may not be entirely surprising, demonstrating it empirically and rigorously is valuable for the community.

- **Methodological rigor**: The experiments are well-designed and carefully controlled, with sound statistical analyses (e.g., mixed-effects models, permutation tests, Bonferroni correction). The use of validated psycholinguistic corpora lends interpretability and psychological grounding to the findings.

- **Comprehensive empirical validation**: The authors examine both real-world open-weight RMs and controlled in-house replications, providing converging evidence for the bias inheritance effect.

**Weaknesses:**

- **Potential domain mismatch**: The preference datasets used for RM training (e.g., HelpSteer, UltraFeedback, HH-RLHF, Argilla-Math) are focused on instruction-following, helpfulness, honesty, and truthfulness, not the kinds of moral or social values represented in the psycholinguistic test sets. Thus, it is not clear whether the persistence of biases reflects insufficient training data volume or an out-of-distribution (OOD) evaluation setting.
- **Formatting issue**: I believe the font used in the submission violates the ICLR template. Please revise this in the updated pdf.

**Questions:**

1. Could the observed persistence of biases be primarily due to the psycholinguistic test corpora being OOD relative to the RM training data? For example, do preference gaps narrow more substantially on in-distribution prompts aligned with RM training datasets (e.g., helpfulness, truthfulness, safety)?

---

Overall, I find this an impactful and a well-executed study. I am willing to increase my score if the authors address my question regarding the OOD nature of the test sets relative to the RM training data.

---

> ### Author Response · Authors · 2025-11-21
> **Response to Reviewer ffKc**
>
> We thank Reviewer ffKc for engaging thoughtfully with our paper.
>
> # D1. The font is nonstandard.
>
> **Action D1.1:** **Resolved** in advance of camera-ready.
>
> # D2. Could the observed persistence of biases be primarily due to the psycholinguistic test corpora being OOD relative to RM training data? E.g., do preference gaps narrow more on in-distribution prompts aligned with RM training datasets (helpfulness, truthfulness, safety)?
>
> **We do not believe that our corpora are as OOD relative to the RM training data as it may appear.**
>
> While our prompts differ stylistically from typical instruction-tuning datasets, namely in their brevity, we do not believe that our results are primarily driven by an OOD evaluation mismatch. We offer three supporting points:
>
> **1. Qualitative prevalence in preference datasets**
>
> **Prompts about variations of the “best thing ever” appear regularly in instruction-tuning and preference datasets**.
>
> For instance, HH-RLHF includes prompts like “What are some of the best book series of all time?” and “What are some of the best sitcoms of all time?” While these are within-category comparisons, they explicitly set up comparative value judgments. HH-RLHF also includes more general statements like “that’s the most important thing to feel happy and secure,” “one of the worst things for some people is the embarrassment they feel when they’re poor,” and “Moms are great because they do everything for you and they’re the best thing in the world.”
>
> UltraFeedback also contains mentions of superlative value judgments, like a statement that meditation “is one of the best things you ever do for yourself,” that encouraging curiosity in children is “one of the best things we can do for our future,” and that “Sometimes, the scariest thing in the world is not knowing.”
>
> **2. Quantitative word-frequency analysis**
>
> We can also quantify the prevalence of relevant data via word frequencies – for instance, here are some relevant word frequencies in Skywork-Reward-Preference-80K-v0.2:
>
> love: 7,070
> freedom: 3,025
> best: 20,466
> greatest: 4,768
> worst: 496
> information: 36,793
> knowledge: 10,932
> family: 8,616
> friendship: 728
>
> (Mean word frequency is 34.3, median is 1, and maximum is 3,059,535 for “the”.)
>
> The HH-RLHF helpfulness dataset has 6,378 mentions of “love”, 5,206 of variations on “free”, etc.; UltraFeedback is similar.
>
> **3. Explicit value tradeoffs in training/preference data**
>
> **Some of the preference data used in top RewardBench RMs as well as the ones we trained in-house directly juxtaposes Agency and Communion values** (e.g., “freedom” and “love”). For instance, one prompt in Skywork includes the following: “I come from a large Italian family who expects me to eventually settle down, marry, and produce plenty of children. But personally, I've only recently realized that none of those expectations align with my dreams. I've always been taught to put family's wishes above my own desires, and for the longest time, I tried to convince myself that I could reconcile those differences. But the truth is... I've never felt the need for a partner or a nuclear family structure. Being alone, or making my own way in life brings me joy.”
>
> We also see Agency/Communion values juxtaposed both cross-culturally in UltraFeedback (“Western philosophy places a strong emphasis on individualism, personal freedom, and autonomy. Eastern philosophy \[places\] greater emphasis on the importance of relationships and community.”) and within-individuals in HH-RLHF (where a user debates whether to move closer to their family: “Assistant: What do you value most? / Human: I think I most value my independence and freedom.”).
>
> Taken together, we believe these provide strong evidence that both the prompts we use and the set of agency/communion terms we highlight are not as OOD with respect to the training data as it might appear.
>
> Reviewer ffKc's question does raise an important point about potential interventions during RM training, not only at pretraining. Our analysis in Fig. 3b shows that, with enough preference data and training time, one can shrink the base-model gap between Agency and Communion rewards. (This also provides evidence that these concepts are indeed present in the preference data.)
>
> Our key contribution is demonstrating that without explicit intervention, values will leak from base models into downstream RMs. Thus one cannot ignore alignment considerations in pretraining, even if there are also actions that can be taken at the RM training stage to mitigate these issues. Further research is needed to characterize precisely how much and what type of preference data would be required to eliminate such biases more systematically.
>
> **Action D2.1:** We will add text to our limitations and discussion section addressing how the composition of training data at different stages of the RLHF pipeline affects the pretraining biases we observe, and highlighting this as a promising direction for future research.

---

### Official Review · Reviewer_8kSZ · 2025-10-30

**Soundness:** 2
**Presentation:** 2
**Contribution:** 1
**Rating:** 2
**Confidence:** 3

**Summary:**

This paper investigates how reward models (RMs) inherit value biases from the pretrained LLMs they are initialized from. Using validated psycholinguistic corpora ("Big Two" and MFD2), the paper finds Llama-based RMs consistently show a preference for "agency" values (e.g., freedom, success, ability), while Gemma-based RMs show a preference for "communion" values (e.g., love, family, friendship). This bias is traced directly back to the log probabilities of the instruction-tuned and even the original pre-trained models. The paper also shows that while the bias in RMs can be mitigated with sufficient preference data, it does not fully disappear.

**Strengths:**

**S1.** The paper clearly traces moral-value biases (agency vs. communion) from the outputs of trained RMs back to the log-probabilities of the base pre-trained models, which provides a clear takeaway that the choice of base model for RM training is also a critical decision that will have downstream value implications.

**S2.** The paper evaluates multiple open-weight RMs based on psycholinguistic validation through controlled ablations on data and base model selections.

**Weaknesses:**

**W1.** The central claim that reward models (RMs) inherit biases from their base pretrained LLMs already feels intuitive and largely expected, given prior research demonstrating bias propagation across fine-tuning and alignment stages [1, 2]. Therefore, this makes the contribution of the paper primarily observational since it does not provide mechanistic interpretability, analysis of latent representations, or deeper causal insight into why such biases emerge. Furthermore, it doesn't offer any actionable mitigation strategy beyond suggesting careful base model selection.

**W2.** The analysis is limited to a narrow experimental scope, focusing only on two base model families (Llama and Gemma; three if we consider Qwen in the Appendix) with two parameter scales (2B and 3B with LoRA), and primarily the agency/communion axis from the Big-Two framework despite referencing MFD2.

## References

[1] Fulay, S., Brannon, W., Mohanty, S., Overney, C., Poole-Dayan, E., Roy, D., & Kabbara, J. (2024). On the relationship between truth and political bias in language models. arXiv preprint arXiv:2409.05283.

[2] Xiao, J., Li, Z., Xie, X., Getzen, E., Fang, C., Long, Q., & Su, W. J. (2025). On the algorithmic bias of aligning large language models with rlhf: Preference collapse and matching regularization. Journal of the American Statistical Association, (just-accepted), 1-21.

**Questions:**

In terms of experimentation, it may be interesting to model scaling law behavior rather than just observe it, such as how increasing preference data or compute (model size) affects the persistence of inherited biases, along with proposing a concrete methodology for mitigating such biases at larger scales.

---

> ### Author Response · Authors · 2025-11-20
> **Response to Reviewer 8kSZ**
>
> We thank Reviewer 8kSZ for the careful reading and encouraging feedback on our “clear takeaway” and “controlled ablations.”
>
> # C1. Is it expected that RMs inherit value biases from pretraining, given Fulay et al. (2024) and Xiao et al. (2025)?
>
> **Our work builds directly on Fulay et al. and Xiao et al., resolving the open question from Fulay et al. and showing that despite important theoretical advances from Xiao et al., the regularizer alone cannot be relied upon to mitigate bias from pretraining.**
>
> Fulay et al. observe evidence of political bias in both truthfulness-trained and control RMs but leave the source of the bias as an open question, concluding: “Identifying the source of this bias is a promising direction for future research.” Our work resolves precisely this question, identifying base models themselves as the ultimate source of downstream RM bias.
>
> Xiao et al. show that bias propagates through KL-regularization, however our work reveals their PM regularizer only solves half the problem. RMs themselves inherit biases from pretraining, so PPO faces bias through both the regularizer *and* the reward.
>
> As Reviewer ffKc argues, “Even though the result may not be entirely surprising, demonstrating it empirically and rigorously is valuable for the community.”
>
> **Action C1.1:** We will cite Fulay et al. (2024) and Xiao et al. (2025) in §5, clarifying how we directly extend their results.
>
> # C2. On mechanistic interpretability and evaluation methods:
>
> **Mechanistic interpretability is an interesting direction for future work, however the paper’s core empirical findings and methodological contribution stand on their own.**
>
> We trace biases from RM outputs to the logprobs of instruction-tuned pretrained base models, establishing a clear causal chain across the entire training pipeline. Pretraining data for these models is not publicly available, making it impossible to use influence functions to trace the origins of these biases further upstream into pretraining data, and the practical implication for model developers would remain unchanged by identifying (e.g.) which attention heads encode these preferences.
>
> Our methodological contribution demonstrates how validated psycholinguistic instruments can be productively adapted to AI evaluation. As Reviewer 8kSZ notes, providing evals with “psycholinguistic validation” constitutes one of the core contributions of the paper. It is our hope that our work encourages more systematic adoption of validated human-subjects measurement tools in alignment research.
>
> # C3. Is the model family/size scope too narrow?
>
> **The models we explore represent a vast majority of open-weight model market share and span two orders of magnitude of model size.**
>
> Llama, Gemma, and Qwen comprise 100% of top-20 open-weight RewardBench RMs and 75% of open-weight market share.
>
> While §4 focuses on 2B and 3B models for compute reasons, **80% of real-world RMs we evaluate in §2 are 8B or 27B.**
>
> Table 1 shows the MWLR extrema from Llama-3.2-3B to Gemma-2-2B (“Freedom” to “Love”); this replicates for Gemma-2 9B and 27B.
>
> Across all instruction-tuned Gemma 2 (2-27B) and Llama 3.x (1-70B) models, the **MWLR score for “Freedom” is *always* greater than “Love.” This gap increases with model size:** see https://imgur.com/a/omPQuHh.
>
> **Action C3.1:** We will include this figure in §3, the 9B-27B MWLR score tables in our supplement, and clarify that this effect appears robust up to 70B.
>
> # C4. Is the Big-Two scope too narrow?
>
> Our purpose is to show that **RMs inherit value biases from pretraining**, and to offer a **methodological approach from psycholinguistics** that can be applied in the wider community for subsequent analyses.
>
> We demonstrate that validated psycholinguistic instruments reveal systematic value differences between model families, that these differences originate in pretraining and propagate through reward modeling, and that base model choice matters for value alignment, not just performance. The Big Two are among the most fundamental constructs in psychology; we welcome future work applying validated psycholinguistic AI evals to other value dimensions.
>
> # C5. It would be valuable to derive scaling laws and concrete mitigation methodologies.
>
> Our results span 2 orders of magnitude of both preference data and model size, and **deriving scaling laws for this effect is a key direction for future work.**
>
> Mitigation approaches are a growing literature, including data filtering, reweighting, augmentation, and debiasing. Our core contribution is to provide developers with clear empirical evidence that such mitigations are essential.
>
> **Establishing a problem’s source is a necessary prerequisite to solving it.** The field cannot develop targeted interventions for base-model value bias until we’ve rigorously demonstrated it occurs, characterized its magnitude, and traced its origins. Our work provides this foundation.
>
> **Action C5.1:** We will note these as vital future work in §6.

---

> > ### Comment · Reviewer_gN69 · 2025-11-24
> >
> > I would second ffKc: In my opinion rigorously test the bias is itself a contribution --- even if everyone is expecting such bias it is necessary for someone to perform the rigorous tests and quantify them, with clear documented methods and conclusions so that the results are no longer folklore and we have a common ground to move forward. I personally appreciate the authors for taking up this responsibility and the results should be out.
> >
> > I think the work compliments Xiao et al. 2025 in the sense that Xiao et al. 2025 showed propagation through RLHF, this work showed RM training itself can be biased. If we think the alignment process as distilling human preference to LLM, with RM this is a two step process, i.e., Human-> RM; RM-> LLM. The authors' work is on the first step while reminded us that that it is not simply Human->RM, but (Human, pretrained-LLM)->RM and the pretraining itself biases RM. This in my opinion complementary to Xiao et al. 2025 which focused on second part of the process, RM->LLM with e.g., PPO.

---

> > ### Comment · Reviewer_8kSZ · 2025-11-26
> >
> > Thank you to the authors and Reviewer gN69 for the detailed responses and suggestions. While I appreciate the effort to provide additional evidence regarding value biases, and I do believe some community members will find this interesting, I would like to maintain my stance that rigorous testing and clear quantification are baseline requirements for scientific publication, not a "contribution" in and of themselves. I strongly believe that in a venue like ICLR, sound methodology and transparent reporting are the expected standard, not a distinguishing novelty.
> >
> > Similarly, the strengths highlighted by other reviewers largely focus on methodological rigor or alternative framing rather than fundamental novelty. In fact, there appears to be a consensus shared by the authors and acknowledged by fellow reviewers that the findings, while carefully presented, are not particularly unexpected. While empirical validation has value, as per my suggestions, the paper doesn't offer additional significant insights or actionable mitigation strategies.
> >
> > Therefore, based on all conversations and in the context of other competitive ICLR submissions, unfortunately, I would like to maintain my score.

---

> > > ### Author Response · Authors · 2025-12-03
> > >
> > > We thank Reviewer 8kSZ for the continued engagement, and Reviewer gN69 for underscoring the value of our contribution. We respectfully disagree with Reviewer 8kSZ and would like to highlight several points:
> > >
> > > * **This result does not exist in the literature.** Many results can be thought of as arguably intuitive with the benefit of hindsight, but the fact that reward models inherit significant and persistent value biases from pretraining is a serious issue with clear significance for model developers. There is no prior work demonstrating this.
> > > * **Making a result quantitative, robust, and rigorous is valuable.** Whether a result confirms one’s priors or not, priors (or even anecdotal wisdom) are not a sufficient bar for scientific evidence. Our contribution is valuable in quantifying this effect and rigorously demonstrating it under controlled conditions and using a variety of different methods. This work is a necessary prerequisite for mitigation.
> > > * **We discuss concrete mitigations:** (1) increasing preference-data size and degree of regularization, which we explore directly in §4; and (2) training data filtering, which is not possible with the model families we study but as we note in §6 comprises valuable future work.
> > > * **Our work *directly* addresses the unanswered questions** from the publications suggested by Reviewer 8kSZ. Fulay et al. observe evidence of political bias in RMs, not only in truthfulness-trained RMs but also in control (“vanilla”) RMs. They suggest that these biases “could be…introduced in reward-model training.” However, the authors find no correlation in all but one of their preference datasets (pruning political content in the truthfulness datasets and looking for correlations with stylistic artifacts). The result is something of a mystery. *Does* this bias get introduced during reward modeling? And if so, where does it come from? The authors write in their conclusion, “Identifying the source of this bias is a promising direction for future research.” The work presented here addresses *precisely* this question. We identify bias patterns in real-world RMs as well as a diverse set of RMs trained in-house on numerous sizes and sources of preference data, and trace these patterns all the way into the instruction-tuned and pre-trained log probabilities – and, importantly, *not* to RM training as Fulay et al. suggest. (In fact we show that pretraining bias can be partially *mitigated* by preference training.) MWLR-score analysis reveals that pretraining bias is a model-family-level effect, visible *before any preference training takes place* – not only in the logprobs of the base models but in the implicit RM between them – and persisting stably across four minor version releases and two orders of magnitude of model size.

---

### Official Review · Reviewer_XumC · 2025-10-31

**Soundness:** 3
**Presentation:** 4
**Contribution:** 4
**Rating:** 6
**Confidence:** 4

**Summary:**

This paper presents case studies that investigate how reward models (RMs) inherit systematic value biases from the base LLMs on which these RMs are instantiated. The authors examined the preferences on a token level along dimensions characterised by two psycholinguistic corpora (Big Two - Agency vs. Communion, and Moral Foundations Dictionary - 5 further aspects). By analysing token preference differences on 10 open-source RMs, the authors validate that Gemma- and Llama-based RMs have systematic differences. Further investigations on the Big Two corpora shows the differences can be traced back to the token probabilities of pre-trained Gemma and Llama models. Experiments tracking Big Two token preferences during RM training show that the inherited biases persist.

These insightful findings draw attention to the choice of base LLMs for RM training and thus the entire LLM pipelines, urging the research community to reflect on current standard practices.

**Strengths:**

- This paper offers some fundamental insights for the research community on choosing base LLMs for RM training, which is underexplored. A focus shift from pure performance metrics to more fine-grained details on value biases is much needed these days.
- The investigations done in the paper make sense and are quite novel, providing solid evidence of the inheritance traces of value biases. Experiments also cover diverse aspects.
- Clarity is excellent - clear motivation, adequate and in-depth discussions, good coverage of related work, and discussions on limitations.

**Weaknesses:**

- The RMs used in Sections 3 and 4 are quite small (2B and 3B), somewhat limiting the significance of results and the validity of relevant claims.
- Sections 3 and 4 focus on a binary value distinction between "Agency" and "Communion". This seems a bit arbitrary. It is also obvious that different types of LLMs (Llama vs. Gemma) would have systematic differences. I would assume that if I randomly choose two common value aspects to repeat the same investigations, I would observe different preferences anyway. Could the authors comment on this choice?
- I have the impression that the findings this paper presents are an instantiation of a common phenomenon, model multiplicity, that we can obtain machine learning models that perform similarly but differ in their internals for the same task, in the realm of reward modelling and LLM training. How is your finding different from something like, "for a tabular classification task predicting loan default, I find one neural network prefers feature AGE more, and another neural network prefers feature LOAN AMOUNT more"?

**Questions:**

See weaknesses.
- One additional comment: there are also RMs that are trained to explicitly predict scores along certain axes (helpfulness, verbosity, coherence, etc., see datasets like HelpSteer) in a multi-regression style. These prediction signals could potentially be a better playground to perform your investigations.

---

> ### Author Response · Authors · 2025-11-20
> **Response to Reviewer XumC**
>
> Thank you for the thorough reading of our work. We appreciate the estimation that it offers “fundamental insights for the research community,” that its investigations “are quite novel, providing solid evidence,” and that its “clarity is excellent.”
>
> # B1. Are the significance of results limited by training 2-3B models?
>
> **Analysis of real-world RMs from 2B to 27B and implicit RMs from 1B to 70B replicate our key results. Effects appear to *increase* with model size.**
>
> Our in-house RM training in §4 focuses on 2B and 3B models for compute reasons, however in our analysis of real-world RMs in §2, **80% of real-world RMs we evaluate are 8B or 27B.**
>
> In §3 **we look at implicit RMs between base models ranging from 1B to 70B**. Table 1 shows the MWLR score from Llama-3.2-3B to Gemma-2-2B, but the score to Gemma-2-9B also goes from “Freedom” to “Love”. So does Gemma-2-27B.
>
> Comparing *all* instruction-tuned Gemma 2 models (2-27B) and all instruction-tuned Llama 3.x models 1-70B, the MWLR score for “Freedom” is *always* greater than “Love” (see https://imgur.com/a/omPQuHh).
>
> Across these 21 model comparisons, “Freedom” ranks among the highest in 17, while “Love” ranks in the bottom two in all 21.
>
> Notably, the MWLR gap between “Love” and “Freedom” *increases with Gemma-model size* for any given Llama model, and (with a single exception) *increases with Llama-model size* for any given Gemma model.
>
> **Action B1.1:** We will include the above figure in our main text and additional MWLR tables in our supplement, clarifying this effect appears to be robust (indeed, increasing) up to 70B.
>
> # B2. Is Agency/Communion a somewhat arbitrary choice?
>
> **These axes are motivated by prior empirical work and constitute two of the most core attributes in social psychology.**
>
> Our work was inspired by Christian et al. (2025), who observed that a Llama RM rewarded “Freedom” most highly, while a Gemma RM trained with the same data rewarded “Love.” We use the Big Two to rigorously evaluate this as evidence of a larger phenomenon.
>
> These axes are not merely convenient for distinguishing “Freedom” from “Love”; rather, they represent two of the most fundamental attributes in social psychology, reflecting essential human aims (goal achievement and meaningful relationships), key personality dimensions and human values, and the most frequent themes in memories, descriptions of self and others, and perception of groups  (Pietraszkiewicz et al. 2019).
>
> **They show a clear “double-dissociation”:** for positively framed prompts, Llama RMs score agency terms higher than Gemma RMs, while Llama RMs score communion terms lower than Gemma RMs. For negatively-framed prompts, Llama RMs score agency terms *lower* than Gemma RMs and score communion terms *higher*. This double-dissociation is strong evidence these are meaningful, coherent axes of difference.
>
> # B3. If someone randomly chose two values, would they observe different preferences?
>
> **The clean double dissociation on Agency/Communion suggests a deeper systematic difference between model families, not arbitrary variation.**
>
> The MFD analysis in Fig 1B shows that despite variability in other dimensions, the completely clean double-dissociation we observe with Agency and Communion is rare. While 9 of 10 prompt/framing pairs show significant differences, only Care shows a significant double-dissociation (consistent with Communion, which overlaps with Care terms).
>
> # B4. Can these findings be understood as model multiplicity?
>
> **Our findings go beyond model multiplicity in important ways.** Unlike idiosyncratic feature preferences in tabular classifiers, which vary with random seeds, we show:
>
> (1) **Systematic family-level differences across minor versions and two orders of magnitude in size:** Llama-3.x RMs in the wild (3-8B) prefer Agency relative to Gemma RMs, while Gemma-2 RMs (2-27B) prefer Communion relative to Llama RMs. This generalizes to MWLR scores across all combinations of Llama-3.x (1-70B) and Gemma-2 (2-27B).
>
> (2) **Persistence despite alignment efforts:** Training our own RMs shows persistent difference between model families despite ablations of data source and quantity, suggesting these representations are deeply rooted and resistant to alignment.
>
> **Action B4.1:** We will cite Black et al. (2022) on model multiplicity and add discussion distinguishing our findings – showing base-model family has systematic, persistent effects.
>
> # B5. Might it be fruitful to explore multi-objective RMs?
>
> **In §2 we examine several real-world RMs that are multi-objective:** QRM-Gemma-2-27B (rank 3 on RewardBench at time of submission) and QRM-Llama3.1-8B (rank 12) are Quantile Reward Models (Dorka 2024) that aggregate distributions over helpfulness, harmlessness, etc. **These show the same base-model effects, suggesting architectural choices alone may not eliminate pretraining bias.** Further exploring diverse reward-model architectures is an interesting avenue for future work.

---

### Official Review · Reviewer_gN69 · 2025-10-31

**Soundness:** 3
**Presentation:** 3
**Contribution:** 3
**Rating:** 8
**Confidence:** 3

**Summary:**

The authors studied the problem of how base model induces inherent bias in the reward model that was fine tuned from them, using methods from psycholinguistics. Using two existing psycholinguistics datasets from domain experts, the authors tested RM from several based models and concluded they indeed induce inherent biases. They then traced the source of the biases and find log probability can explain these biases from the base model.

**Strengths:**

- The authors took an interdisciplinary approach in viewing the question --- we should indeed borrow existing domain knowledge from disciplines that studies human values.
- The statistical tests are careful in e.g., FDR control
- The framing of model difference as reward difference is an interesting view point

**Weaknesses:**

Exhaustive token search might itself have limitations, I am not sure how the prompting scheme have an influence in this process.

**Questions:**

## Statistical analysis
- The difference can be significant but not large, and a small p-value can be due to large sample size. I am not very sure what is the best way to interpret the differences the authors reported in Fig.1, they do not seem to be large in some cases. Especially since the authors showed error bar using standard errors rather than standard deviations. I have no doubt the differences are *statistically significant*, but are they really *meaningful*? E.g., median rank of 1000 and 1001 might not be that meaningful even if the difference is significant because the estimation error is small.

-  Fig.2 provides a bit more insight since the density plot.

## Implicit reward

- Is it fair to say that the author would also view KL between two models to be reward that can make model A to B if trained with RL?

## Prompting
- Does prompting have an influence in vocab search?

---

> ### Author Response · Authors · 2025-11-20
> **Response to Reviewer gN69**
>
> Thank you for your careful reading of our work.
>
> # A1. Exhaustive token search may have limitations. How does the prompting scheme influence the process?
>
> Exhaustive search surfaces provably optimal/pessimal responses within a given length and avoids the need for sampling (and choice of temperature and sampling algorithm) as in more generative forms of evaluation. **Acknowledged weaknesses of exhaustive token search include:** (i) Short responses restrict the scope of prompts that can be studied. (ii) Focusing on the first response token excludes inference-time compute (CoT). (iii) It requires care when comparing across different tokenizers.
>
> However, **empirical evidence supports exhaustive token search:**
>
> First, **results generalize across prompt perturbations:** we expanded the 3 prompts used by Christian et al. (2025) to 54 prompts, and find that the effects we observe generalize across prompt variations.
>
> Second, **multi-token responses replicate single-token results**. We note that Christian et al. (2025), which introduced exhaustive token search, used techniques from the jailbreaking community such as Greedy Coordinate Gradient (GCG) to derive *near*\-optimal model responses at greater lengths (2-token, 9-token, etc.). These reproduce the same qualitative patterns observed in the provably optimal/pessimal single-token responses, offering preliminary evidence that single-token findings generalize to longer sequences.
>
> **Action A1.1:** We will clarify both weaknesses and strengths of exhaustive token search in the limitations section.
>
> # A2. The effects you observe are significant – but what is their *size*?
>
> **Statistical analysis shows a “medium” effect size.** The average movement in the *median* rank for Big-2 nouns in Gemma vs Llama RMs, reported in Fig. 1A, is approximately 5/263 ranks ($-4.21$ for Agency and $+4.58$ for Communion). This effect corresponds to approximately half of the pooled standard deviation of median ranks across models and prompts (10.7 across all positive prompt variants) or a Cohen's *d* of .40-.43, traditionally considered a medium effect size.
>
> **Action A2.1:** Median-rank and Cohen’s *d* analyses will be added to §2.
>
> **Top-*k* analysis suggests meaningful differences for downstream LLM policies.** One can also frame effect size in terms of how differences in RMs shape downstream LLM behavior. Thus we can ask how bias manifests in the *highest* scoring tokens for Gemma vs Llama-based RMs, since these will be most reflected in a finetuned LLM’s policy.
>
> Importantly, when we focus on the full exhaustive token search (across our 10 leading RewardBench RMs) and analyze the top 10 scoring tokens, we find that for the Gemma RMs, on average 5 are Communion tokens (e.g. “love,” “compassion,” “harmony”) and 0 are Agency – whereas for Llama, on average 3.67 are Communion tokens and 2.33 are Agency (e.g. “freedom,” “opportunity”). Communion tokens rank 2.88 (of 10\) for Gemma, and 3.75 (of 10\) for Llama; by contrast Agency has no rank for Gemma (since it doesn’t figure in the top 10 tokens) and an average rank 6.67 (of 10\) for Llama.
>
> All in all, this suggests that these biases manifest in meaningful ways in RM reward scores, as well as in the downstream LLMs that optimize for them.
>
> **Action A2.2:** Top-*k* analysis will be added to §2.
>
> # A3. Error bars in Figure 1 are presented using standard errors and bee-swarm plots. Can we see standard deviations and density plots instead?
>
> **Action A3.1:** A high-resolution version of Figure 1a with both standard deviations and violin density plots is available at [https://imgur.com/a/qhYSNtG](https://imgur.com/a/qhYSNtG). This figure will be added to the paper supplement.
>
> **Action A3.2:** Figure 1a in the main text will also now include density plots for clarity.
>
> # A4. Do the authors also view KL between two models to be an implicit reward that can make model A to B if trained with RL?
>
> Yes. **KL and Reverse KL implicit rewards appear in Appendix E**:
>
> The KL integrand is simply $p\_1 \\cdot (\\log p\_1 \- \\log p\_2)$, and so it is a very natural candidate for weighted implicit reward. In Appendix E we present results weighted by both $p\_1$ (KL) and $p\_2$ (reverse KL), as well as a version of Jensen-Shannon divergence, though our analysis favored the sensitivity of the MWLR score.
>
> One nice benefit of the MWLR score is that, unlike KL (where $D\_\\mathrm{KL}(p\\|q) \\neq \-D\_\\mathrm{KL}(q\\|p)$), it is antisymmetric, such that $\\mathrm{MWLR}(p\\|q) \= \-\\mathrm{MWLR}(q\\|p)$. In Table 1, $\\mathrm{MWLR}(\\texttt{Llama-3.2-3B-Instruct}\\|\\texttt{gemma-2-2b-it})$ produces the “Freedom–Love” axis; antisymmetry means that reversing the model order would produce a “Love–Freedom” direction, with the terms in reverse order and the signs of the scores flipped.
>
> **Question A4.1:** We will expand Appendix E to include additional discussion of KL-based implicit rewards and their relationship to MWLR, including the antisymmetry property.

---

> > ### Comment · Reviewer_gN69 · 2025-11-24
> >
> > I appreciate the authors response! Please add these to the revised paper and my assessment to this work remain positive.

---

### Comment · Area_Chair_7BbF · 2025-11-25
**Please revise your paper accordingly**

This is a notice for the authors to revise the paper now rather than waiting for the camera-ready stage. I will prepare the meta-review based on the final revised version. Please incorporate the requested changes during this rebuttal period.

Additionally, please update the font to comply with the original ICLR formatting requirements before I proceed with the necessary actions. Using a different font can affect the paper length and may result in a desk rejection.

Thank you.

---

> ### Author Response · Authors · 2025-11-26
> **Response to Area Chair: Paper Revised**
>
> Dear Area Chair,
>
> Thank you for the guidance. We have uploaded a revised manuscript incorporating all requested changes from the reviewers, including:
>
> - Expanded discussion of exhaustive token search limitations and strengths (§6)
> - Cohen's _d_ effect sizes and top-_k_ analysis (§2)
> - Violin density plots in main text and supplement (Fig. 1, Fig. A1)
> - Extended discussion of MWLR and KL-based implicit rewards (Appendix F)
> - Additional MWLR figure showing effect robustness to model size (§3), and tables for 9B and 27B models (Appendix F)
> - Citations to Fulay et al. (2024) and Xiao et al. (2025) with discussion of how our work extends theirs (§5)
> - Discussion of model multiplicity (§5)
> - Discussion of scaling laws, mitigation strategies, and training data composition as future work (§6)
>
> Regarding the font: loss of Times New Roman was an unintended consequence of switching from pdflatex to xelatex for Chinese character support in Table 1. This has been corrected in the revised submission, and the paper is still within length requirements.
>
> Thank you for your attention to our paper.
>
> Best regards,
> [Authors]

---

### Author Response · Authors · 2025-12-03
**Closing Statement**

Dear Area Chair,

We wish to thank you, the reviewers, and the previous area chair for helpful feedback that has improved this paper significantly. We are pleased that our initial review scores were strong (8, 8, 6, 2). Below we address the outlier score and the concrete improvements to the paper.

Reviewer 8kSZ maintains that our findings are “not particularly unexpected.” We respectfully note that Fulay et al. (2024), one of the two papers cited by the reviewer, observes apparent political bias in RMs but concludes that “Identifying the source of this bias is a promising direction for future research.” Our work directly resolves this question, by tracing observed value differences in RMs all the way into the log probabilities of their instruction-tuned and pretrained base models. Establishing a phenomenon's existence, magnitude and robustness, and pinpointing its causal origin, is a necessary prerequisite to developing targeted interventions. Reviewers ffKc and gN69 both articulate this view, with Reviewer gN69 “seconding” Reviewer ffKc in noting that “it is necessary for someone to perform the rigorous tests and quantify them... so that the results are no longer folklore.”

We thus are confident the majority consensus among reviewers is that this work provides considerable value for the wider community. However, we have also made significant additions since our initial submission following queries and suggestions from all four reviewers, which have strengthened the manuscript further:

1. **Effect size clarifications (§2):** Cohen's *d* analysis and top-*k* analysis now quantify the magnitude and downstream relevance of observed biases.
2. **Improved visualizations (Fig. 1 \+ Fig. A1):** Violin plots now convey distributional density; standard deviations added in supplement.
3. **Robustness to model scale (§3 \+ Appendix F):** New figure showing MWLR scores for “Love” and “Freedom” across all Gemma 2 and Llama 3.x models from 1B to 70B, detailing that these terms occupy opposite extremes for all implicit Gemma 2 → Llama 3 RMs, with gaps *increasing* as a function of model size. Tables A2 and A3 provide optimal/pessimal tokens for 9B and 27B models. This set of additional analyses directly addresses concerns raised by Reviewer XumC (Score of 6\) and Reviewer 8kSZ (Score of 2).
4. **Extended related work (§5):** Summaries of Fulay et al. (2024) and Xiao et al. (2025) clarify how our work resolves the former's central open question and complements the latter. New subsection on model multiplicity citing Black et al. (2022).
5. **Methodological discussion (§6):** Expanded treatment of token search strengths and limitations; training data composition and mitigation strategies discussed as future work.
6. **Extended technical discussion (Appendix F):** Extended discussion clarifies the relationship between MWLR score and KL-divergence.

We understand that reviewing involves substantial expertise, time, and attention, and we are grateful for input that has made our paper significantly stronger since initial submission.

Best regards,
\[Authors\]

---

### Meta-Review · Area_Chair_GFRf · 2026-01-06

**Summary:**

Most reviewers have provided very positive feedback on this paper. While the reviewer who gave a score of 2 raised some doubts about the paper’s innovativeness, I believe the academic community needs scholars to systematically analyze the impact of pretraining on Reward — so I recommend accepting this paper.

**Reviewer Scores:**

None

---

### Decision · Program_Chairs · 2026-01-26

Accept (Poster)